# Woeseiales transcriptional response to shallow burial in Arctic fjord surface sediment

Joy Buongiorno[1¤]*, Katie Sipes[1], Kenneth Wasmund[2,3], Alexander Loy[2,3], Karen G. Lloyd[1]

1 Department of Microbiology, University of Tennessee, Knoxville, Tennessee, United States of America,
2 Division of Microbial Ecology, Centre for Microbiology and Environmental Systems Science, University of Vienna, Vienna, Austria, 3 Austrian Polar Research Institute, Vienna, Austria

¤ Current address: Division of Natural Sciences, Maryville College, Maryville, Tennessee, United States of America
* joy.buongiorno@maryvillecollege.edu

**Data Availability Statement:** Sequences for 16S rRNA gene libraries, transcriptomes, and metagenomes are deposited at NCBI under BioProject PRJNA493859. MAGs can be accessed from IMG with Gold Analysis Project IDs

## Abstract

Distinct lineages of Gammaproteobacteria clade Woeseiales are globally distributed in marine sediments, based on metagenomic and 16S rRNA gene analysis. Yet little is known about why they are dominant or their ecological role in Arctic fjord sediments, where glacial retreat is rapidly imposing change. This study combined 16S rRNA gene analysis, metagenome-assembled genomes (MAGs), and genome-resolved metatranscriptomics uncovered the *in situ* abundance and transcriptional activity of Woeseiales with burial in four shallow sediment sites of Kongsfjorden and Van Keulenfjorden of Svalbard (79˚N). We present five novel Woeseiales MAGs and show transcriptional evidence for metabolic plasticity during burial, including sulfur oxidation with reverse dissimilatory sulfite reductase (*dsrAB*) down to 4 cm depth and nitrite reduction down to 6 cm depth. A single stress protein, spore protein SP21 (*hsp*A), had a tenfold higher mRNA abundance than any other transcript, and was a hundredfold higher on average than other transcripts. At three out of the four sites, SP21 transcript abundance increased with depth, while total mRNA abundance and richness decreased, indicating a shift in investment from metabolism and other cellular processes to build-up of spore protein SP21. The SP21 gene in MAGs was often flanked by genes involved in membrane-associated stress response. The ability of Woeseiales to shift from sulfur oxidation to nitrite reduction with burial into marine sediments with decreasing access to overlying oxic bottom waters, as well as enter into a dormant state dominated by SP21, may account for its ubiquity and high abundance in marine sediments worldwide, including those of the rapidly shifting Arctic.

## Introduction

Seafloor sediments host a significant portion of Earth's biomass [1]. Microorganisms must have physiological adaptations to be active in marine sediments. Steep gradients in oxygen, redox active compounds, organic matter compositions, metal distributions, and metabolite

Ga0315927, Ga0315928, Ga031529, Ga0315930, and Ga0315931. R Markdown files for Spearman calculations and depth-resolved transcript coverage plots for all individual genes are deposited at GitHub (https://github.com/JBuongio/Woes).

**Funding:** This work was primarily supported by grants from the Simons Foundation (404586 to K. G.L.; https://www.simonsfoundation.org/) and the Austrian Science Fund (P29426-B29 to KW; P25111-B22 to AL; https://www.fwf.ac.at/en/), with additional funding from a student research grant from the Explorer's Club (to J.B.; https://www.explorers.org/) and the U.S. Department of Energy, Office of Science, Office of Biological and Environmental Research, Genomic Science Program under Award Number DE-SC0020369 (to K.G.L.; https://genomicscience.energy.gov/). The funders had no role in the study design, data collection and analysis, decision to publish, or preparation of the manuscript.

**Competing interests:** The authors have declared that no competing interests exist.

pools are some of the many characteristics to which a microorganism must be able to respond [2–4]. Alternatively, increasing energy limitation with depth in the subseafloor environment may promote a strategy of metabolic standby until conditions again become optimal [5, 6], perhaps through exhumation through bioturbation, sediment slumping, or storm events in shallow waters. The particular strategy used by sedimentary microbes will likely depend on the sedimentological regime, degree of bioturbation, and depositional history.

One group that may have adaptive strategies for dealing with diverse sediment environments is the order Woeseiales (formerly 'JTB255') [7]. The Woeseiales are a globally-distributed group of Gammaproteobacteria that have been shown through 16S rRNA gene amplicon sequence surveys, as well as catalyzed reporter deposition fluorescence *in situ* hybridization (CARD-FISH), to be highly abundant in both coastal and deep-sea surface sediments [7–11]. Metagenomic studies [7] and 16S rRNA gene oligotyping studies [12] have revealed that this clade is composed of distinct lineages that inhabit either coastal or deep-sea sediments, suggesting that subclades within Woeseiales may interact with their sediment environment differently. Single-cell amplified genomes (SAGs), metagenome-assembled genomes (MAGs), transcriptomics, and incorporation of $^{13}$C-labeled inorganic carbon showed the potential for chemolithoautotrophy via the oxidation of reduced sulfur compounds and the Calvin Benson cycle in coastal subclades [13]. A stronger potential for heterotrophy via oxidation of carbohydrates, peptides, aromatics [7] and molecular hydrogen [10] has been suggested for deep-sea subclades [12]. Interestingly, the only cultured representative, *Woeseia oceani* XK5, from a coastal sediment sample, appears incapable of autotrophic growth [14]. This supports the idea that different populations of Woeseiales may have fundamentally different physiologies. Even single representatives of Woesiales have been hypothesized to be metabolic generalists, with evidence for a high degree of metabolic flexibility [15].

The physiological responses of individual populations to burial in marine sediments, such as transcriptional regulation and dormancy strategies, remain largely unexamined and may be key to understanding how Woeseiales taxa modulate their behavior in different sediment environments. We hypothesize that a high depth-resolution metatranscriptomic analysis will demonstrate metabolic plasticity as well as strategies for long-term survival and subsistence in the sediment environment. Altogether, this work will shed light on the ability to extrapolate the functions made previously from genomic inferences of Woeseiales.

Woeseiales have also been found in Arctic fjord sediments in Svalbard, Norway [11, 13]. The Arctic is currently experiencing climatic changes from increased temperatures [16], frequency and severity of storm events [17], and glacial retreat [18]. The ability to predict whether members of a globally important microbial taxon such as Woeseiales will be resilient to this change requires understanding the mechanisms by which it responds to the stresses of burial in marine sediments. We therefore examined changes in gene transcript recruitment with depth to MAGs of uncultured Woeseiales over the upper 5 to 6 cm of sediments from two Svalbard fjords: Kongsfjorden and Van Keulenfjorden. Here, glaciers terminate in marine-influenced fjord systems, where high rates of sediment loading (5–10 cm/year [19, 20]) derive directly from glacial run-off. However, as glaciers retreat, this sediment loading may decrease [20, 21]. We examined two sites from Kongsfjorden (sites F and P) and one site from Van Keulenfjorden (site AC) with heavy loading of glacially-derived sediment, as well as another site from Van Keulenfjorden (site AB) with minimal input of glacially-derived sediments. Five novel medium to high quality Woeseiales MAGs were reconstructed from metagenomic sequences from these four sites. Mapping of transcripts to these MAGs demonstrated broad decreases in transcription with depth for the majority of genes, while one gene, *hsp*A (encoding spore protein SP21), comparatively increased transcription with depth by multiple orders of magnitude.

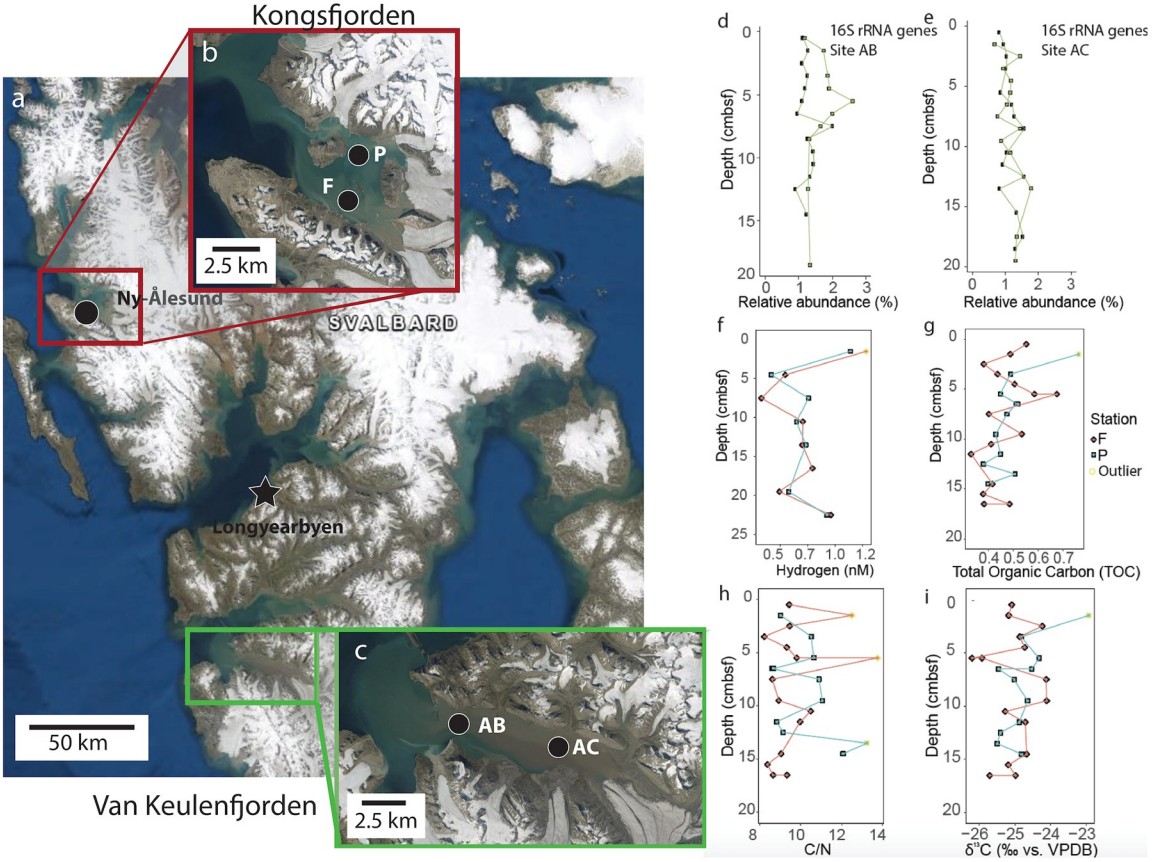

**Fig 1. Physical and geochemical setting.** Overview map of Spitzbergen (a), containing red box indicating Kongsfjorden (b) and green box indicating Van Keulenfjorden (c). Landsat imagery courtesy of NASA Goddard Space Flight Center and U.S. Geological Survey. The relative abundance of Woeseiales by 16S rRNA gene amplicon analysis for sites AB (d) and AC (e) in Van Keulenfjorden are shown along with measurements of hydrogen (f), total organic carbon (g), carbon to nitrogen ratios (h), and stable isotopes of organic carbon (i). Outliers were determined with Cook's distance.

## Results

Sites within Van Keulenfjorden (sites AB and AC) and Kongsfjorden (sites P and F), Svalbard, were investigated for their geochemical properties and relative abundance of Woeseiales with depth (Fig 1a–1c). Previous geochemical analysis on porewaters at these sites showed higher relative concentrations of Fe (up to ~300 nM) at site AB compared to <50 nM at site AC in the top 5 cm of sediment [22]. Within shallow sediments at site AB, Mn concentrations never exceed 65 nM, while at site AC, Mn concentrations exceed 250 nM [22]. Similarly high Fe concentrations (~300 nM) have been observed in shallow sediments at site P in Kongsfjorden, with Mn concentrations not exceeding 20 nM [23].

The relative abundance of Woeseiales 16S rRNA gene amplicons remained steady with depth in Van Keulenfjorden sediment, between 1–3% of 16S rRNA gene amplicon libraries down to ~20 cm at sites AB and AC in Van Keulenfjorden (Fig 1d and 1e). 16S rRNA genes did not amplify for sites P and F in Kongsfjorden, likely due to the high concentrations of Fe, which can interfere with DNA extraction and amplification [24].

In Kongsfjorden, hydrogen, total organic carbon (TOC), C/N ratios, and stable isotopes of organic carbon were variable in the upper five cm (Fig 1f–1i). This is similar to sites AB and

AC in Van Keulenfjorden [22, 25]. High C/N ratios and low TOC values at sites AC, F, and P indicate the deposition of refractory terrestrial organic material relative to site AB where corresponding values were lower [22, 25].

## Recruitment of metagenomic and metatranscriptomic reads to Woeseiales MAGs

Metagenomic sequence libraries were generated from the depth-integrated upper five cm at three of the four sites (AB, AC, and F), with read numbers ranging from 30,615,734 to 83,705,875 per library. The metagenomic sequencing failed for site P, perhaps due to the same coextraction of inhibitors that probably prevented amplification of 16S rRNA genes. Five Woeseiales MAGs were reconstructed from the metagenomic data (S1 Fig) and found to be closely related to the Woeseiales Lineage 11 representative JSS Woes1 [12]. Amino acid identity analysis shows that the majority of proteins in our MAGs match unclassified Gammaproteobacteria and/or Chromatiales (S2 Fig). Less than half of the proteins in the Woeseiales MAGs have average amino acid identity to the proteins from the isolate *Woeseia oceani*, pointing to proteomic novelty in our environmental MAGs compared to the isolate.

MAG completeness ranged from 80 to 94%, with 2 to 9% contamination (S1 Table). Woeseiales collectively recruited from 24% of 6,145,182 reads mapped to the metagenome at site F to 33% of 15,328,660 reads mapped to the metagenome at site AC, highlighting the importance of this clade to the sediment microbial community. Across all transcriptomes (which ranged from 272,522 reads to 1,307,568 reads after removal of rRNA), Woeseiales persistently recruited from 0.5 to 3% of mapped reads. A high similarity in predicted gene functions with KEGG-decoder [26] between the five Woeseiales MAGs permitted a combined analysis for transcripts (S3 Fig).

Most genes that recruited transcripts (54% of total transcribed genes) were recruited only from one of the four sites (S4 and S5 Figs). However, the majority of these transcripts that only appeared at one site (93%) occurred only in a single depth. We therefore restricted analyses to transcripts occurring at more than one depth, to filter out potential noise. This showed transcripts unique to a single site included electron transport proteins, proteins involved in cellular signaling, and chaperones (S5 Fig). Specifically at site AB, we identified transcripts for the oxidoreductase complex protein subunits RnfE (Em = -500 mV [27]) and RsxB, which are involved in electron transfer to regulatory proteins that respond to oxidative stress [28]. At site AC, we identified exclusive transcripts for bifunctional (p)ppGpp synthase/hydrolase SpoT. This enzyme catalyzes the synthesis and degradation of guanosine pentaphosphate ((p)ppGpp), an alarmone involved in signaling and bacterial stringent response to nutritional starvation and the transition to a state of dormancy under stress [29]. At site F, we detected unique transcripts for chaperone ClpB. This chaperone is involved in recovery after heat-induced damage and works in concert with the DnaK chaperone system (transcribed at F, AB, and P) to refold denatured protein aggregates [30]. Finally, at site P, we identified unique transcripts for RNA polymerase sigma factor SigV, an extracytoplasmic function sigma factor involved in transcription under cell envelope stress [31].

Among the genes that were transcribed at all sites (8% of total transcribed genes; S4 Fig), about 20% encoded proteins involved in energy production and conversion (S2 Table). This included transcripts for the central metabolism genes encoding citrate synthase, acetyl-coenzyme A synthase, and isocitrate lyase. In addition, several genes involved in sulfide oxidation metabolism were found at all sites. This included the cytochrome subunit of sulfide dehydrogenase (*fcc*A, present in all MAGs) and the beta subunit of dissimilatory sulfite reductase (*dsr*B). Sequence analysis of the alpha and beta subunits (*dsr*AB) within two Woeseiales MAGs

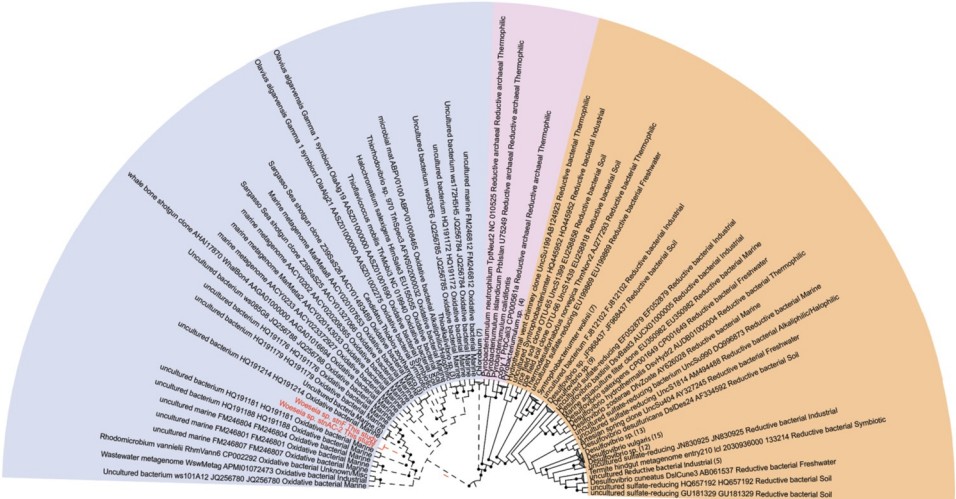

**Fig 2. Phylogeny of DsrA encoded on Woeseiales MAGs.** The CIPRES Science Gateway v. 3.3 online server was used for RAxML and the tree was visualized with the Interactive Tree of Life (iToL) visualization tool. Nodes with bootstrap values equal to or greater than 70% are indicated with a black circle. Reductive-type archaeal and bacterial sequences are represented in pink and orange fields, respectively, with solid tree branches. Oxidative-type bacterial sequences are represented in the blue field with dashed tree branches. Woeseiales DsrA sequences are highlighted in red text.

(Woeseia sp. stnF and Woeseia sp. stnAC-2) confirms that members of this clade encode a reverse dissimilatory sulfite reductase involved in oxidation of sulfur (Fig 2) [7]. In Woeseiales, this may facilitate thiosulfate oxidation through a truncated SOX pathway involving SoxXA, SoxYZ, SoxCD, SoxB and a c-type cytochrome (c-555) (Fig 3a). The oxidation of reduced sulfur and sulfur intermediates is potentially linked to carbon fixation through the Calvin-Benson Cycle using an encoded Type II RubisCO (Fig 3a; S6 Fig; S1 File); however, we were not able to detect transcripts for *sox* genes. Transcription of *dsr* subunits decreased with depth (Fig 3b).

Another gene that was transcribed at all sites was *nir*S. This gene encodes nitrite reductase, a key enzyme in denitrification. Transcripts for *nir*S were detected at every sediment depth (Fig 3b). Similar to r*dsrAB*, transcription of *nir*S decreased with depth. Transcripts of *nir*S were detected in deeper sediment depths compared to transcripts of *dsrAB*, which did not penetrate past 4–5 cmbsf (Fig 3b). While transcription levels of both genes decreased overall, transcription of *nir*S remained elevated below 2 cm, after which *dsr* transcripts were nearly zero at most sites. This may indicate a transcriptional response to decreased availability of respective substrates, with oxygen and intermediates of sulfur becoming depleted at shallower depths relative to nitrite [32–34].

## Transcriptional trends of hspA (SP21)

The gene *hsp*A, encoding SP21, also called heat-shock protein A, recruited at least ten times more transcripts in our Woeseiales MAGs than the next most highly transcribed gene all sites (chaperone protein DnaK, 1,916 TPM; S7 Fig). SP21 is a small, membrane-associated stress protein related to heat shock proteins [35, 36] (S8 Fig). This gene had very high transcriptional coverage, with values an order of magnitude greater on average (TPM = 2,770) compared to all other genes (average TPM = 531). This gene also had the largest transcription observed for any gene at any site, with 309,186 TPM at Site P, 5–6 cmbsf (S7 Fig).

Depth analysis of SP21 coverage with post-hoc Tukey testing showed statistically-significant increases with depth at sites AB and P (ANOVA $p < 0.05$; Fig 4). At sites AC and F,

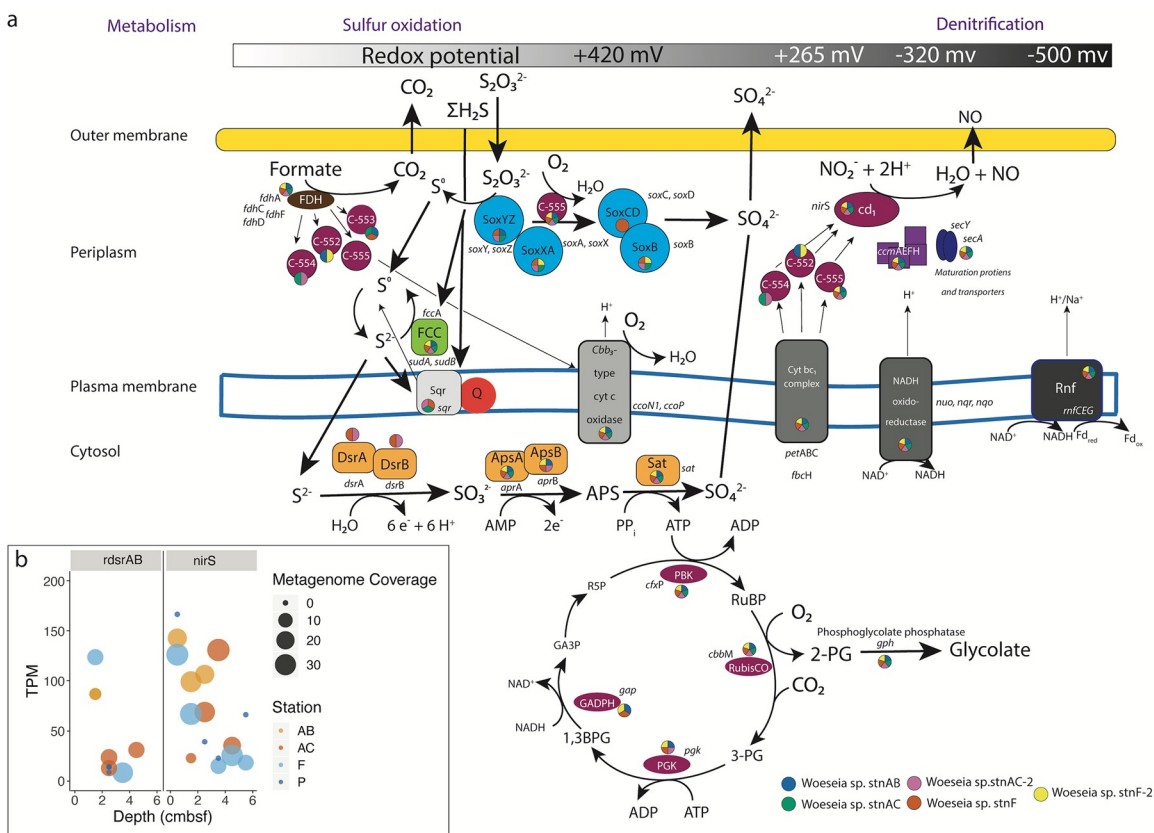

**Fig 3. Sulfur oxidation and denitrification in Woeseiales MAGs.** Schematic representation of metabolic pathways reconstructed in Woeseiales MAGs (a). The gray/black bar at the top corresponds to the redox potential for the key cytochromes depicted in the figure (sulfide quinone reductase, cbb$_3$-type c oxidase, cyt bc1 complex, NADPH oxidoreductase, and the Rnf complex). Colors in circles next to each protein represent the presence in the MAGs. Depth trends in transcriptional coverage (reported in TPM) for *nir*S and the alpha and/or beta subunits of *dsr* are reported (b).

transcriptional coverage did not show statistically-supported trends with depth (ANOVA $p > 0.05$). Instead, at these sites, transcription of spore protein SP21 either remains high at all depths (site F) or in the low-to-middle-range of coverage at all depths (site AC).

In addition to SP21, transcripts for other stress proteins, transcriptional regulators, and starvation genes were detected (S9 Fig). However, transcript recruitment to these genes was much lower than that observed for SP21, at 89 TPM on average. In addition, none of these genes showed increases with depth, even those co-localized on the same contig as the spore protein SP21 gene *hsp*A (S10 Fig). In fact, transcripts for the co-localized chaperonins encoded by *gro*S and *gro*L decreased with depth (S9 Fig), while other chaperones either showed no trend (co-localized chaperon DnaK) or were only detected at the surface (chaperone SurA and chaperone HtpG; S9 Fig).

Other genes flanking the SP21 gene *hsp*A included those encoding the protein-methionine-sulfoxide reductase MsrPQ system (*msr*P and *msr*Q; S10 Fig). This membrane-associated system works with the periplasmic chaperone SurA to repair oxidatively damaged periplasmic proteins containing methionine sulfoxide residues, thereby maintaining cell envelope integrity [37]. Transcripts for chaperone SurA were identified in surface sediments of sites AB and AC (S9 Fig). We identified several additional nearby genes involved in membrane-associated

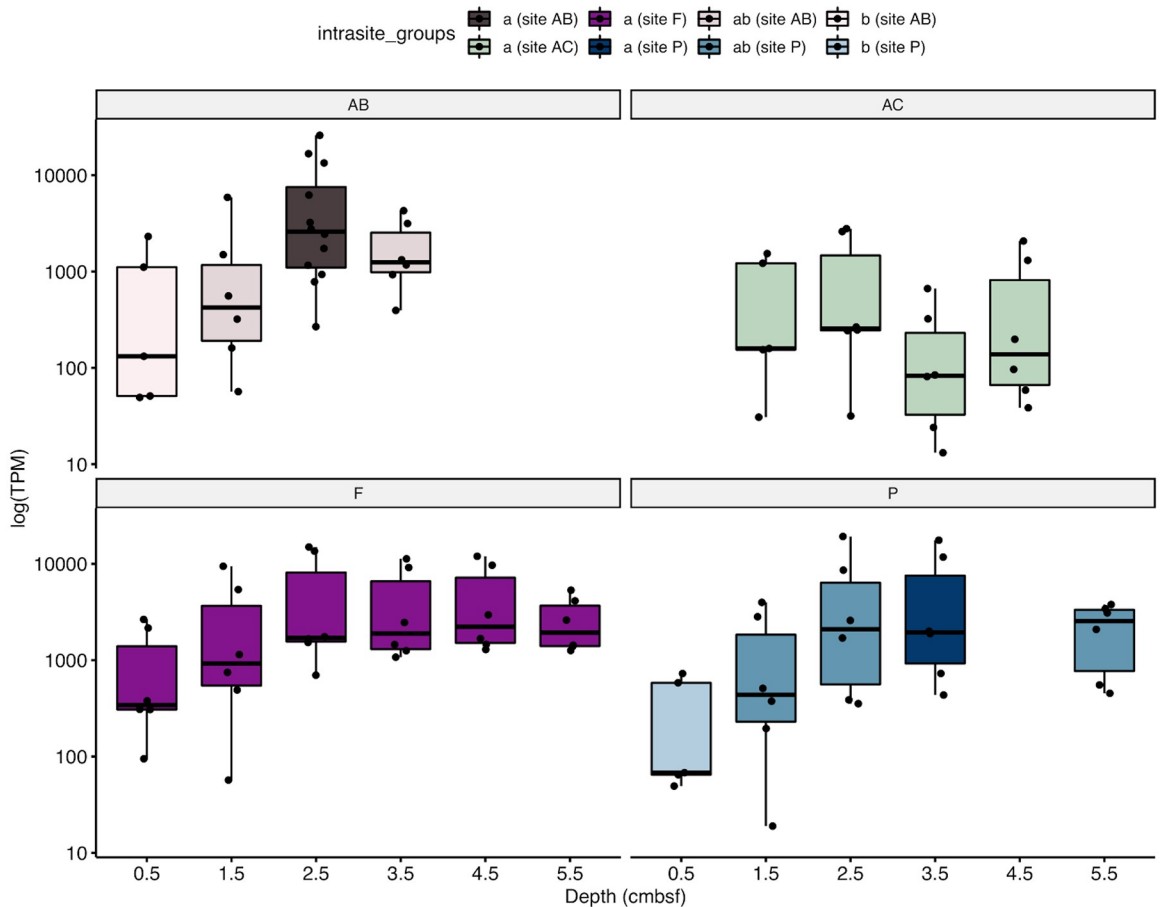

**Fig 4. Transcriptional coverage of SP21 with depth at each site.** Each point represents a copy of the gene encoding SP21 (*hsp*A) in a Woeseiales MAG. Some MAGs have more than one copy of the gene (see S9 Fig). The upper and lower hinges correspond to the 75th and 25th percentiles, respectively, and the median is represented by a horizontal bar. Statistical differences in mean values across depths for a single site was calculated with Tukey means testing. Depths that do not share a letter/color have statistically significant different mean TPM values across the different copies of the gene.

processes. These included genes that participate in toxin removal/efflux (*tol*C transcribed at AB, P, and F; *xcp*T and *eph*A) [38–40], protein repair (*deg*Q transcribed at AC, F, and P; *pcm*) [41, 42], and cell wall reorganization (*upp*P and *glu*P) [43].

## Transcriptional coverage and richness with depth

To get a broader understanding of transcriptional responses to burial, we classified Woeseiales transcripts into Clusters of Orthologous Groups (COGs) (Fig 5). We took this approach because low overall transcriptional coverage and inconsistent detection of transcripts for an individual gene across several depths precluded analysis at the gene-level. Spearman's rho was calculated to assess the degree of association between depth and TPM for each COG category. For this analysis, SP21 was removed to allow equitable comparison.

Information storage and processing genes (mostly involved in transcription/translation) had the highest average transcription amongst the three main COG categories (average TPM = 91) compared to cellular processes and signaling (average TPM = 65) and metabolism (average TPM = 60). With the exception of site P, all sites had significant decreases ($p < 0.05$)

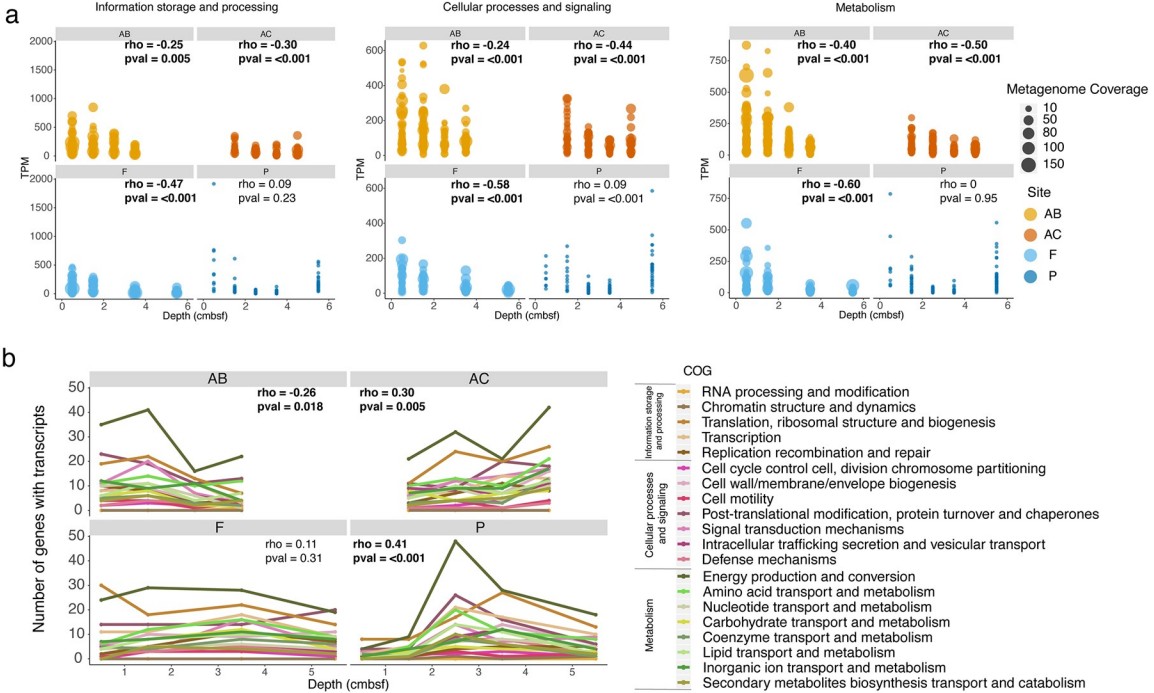

**Fig 5. Coverage trends and transcriptional richness.** Changes in transcription for each of the three main COGs (information storage and processing, cellular processing and signaling, and metabolism) at each site (a). For equitable comparison, SP21 transcripts were removed (because of its anomalously-high coverage values) as well as the two libraries representing the bottom 10% of library sizes (2–3 cm and 4–5 cm at site F). Metagenome coverage is not reported for site P, as we did not sequence a metagenome for this site. Changes in transcriptional richness, or the number of transcribed genes, with depth (b). Lines correspond to a subCOG category within a main category, each rendered to share the same color palette: khaki lines represent information storage and processing transcripts, pink lines represent cellular processes and signaling transcripts, and green lines represent metabolism transcripts. Spearman's rho was calculated for both COG-specific and site-wide trends in transcriptional richness. Only site-wide trends had significant p-values at alpha = 0.05.

in transcripts for all three of these main COG categories (Fig 5a). Metabolism genes had the greatest decrease (Spearman's rho from -0.5 at site AC to -0.6 at site F), followed by cellular processes and signaling (Spearman's rho from -0.24 at site AB to -0.58 at site F) and information storage and processing (Spearman's rho from -0.25 at site AB to -0.47 at site F).

The COG with the largest number of genes that were transcribed was energy production and conversion, followed at most sites by translation, ribosomal structure, and biogenesis (Fig 5b). We tested how richness within these COGs and others changed along a depth gradient through Spearman analysis. This analysis revealed that the number of transcribed genes at site AB decreased overall (Spearman's rho = -0.26, p value = 0.018), while at sites AC and P, there was a slight increase (Spearman's rho = 0.30 and 0.41, respectively, p ≤ 0.005). By contrast, site F had no statistically-supported depth trend in transcriptional richness, either when evaluating the entire data set or sub-setting by COG category (Spearman's rho = 0.11, p value = 0.31).

Combined, these analyses suggest that overall transcription generally decreased, while trends in transcript richness were not uniform across sites. Instead, trends in transcriptional richness broadly intersected with increased SP21 transcription (Figs 4 and 5). For example, at site AB, overall decreases in transcriptional richness coincided with increased SP21 transcription from 0–1 cm to 2–3 cm. At site P, increased richness from 0–1 cm to 2–3 cm was followed by a dramatic decrease from 2–3 cm to 3–4 cm (Fig 5b). This decrease coincided with statistically elevated amounts of SP21 transcription (Fig 4). At site AC, SP21 transcripts remained

low at all depths and were accompanied by an increase in total transcript richness. Interestingly, at site F, we observed no change in either transcriptional richness or in SP21 transcription. Altogether, this suggests that mRNA generation may be attenuated once a threshold of SP21 transcription is reached.

## Discussion

Woeseiales populations made up 1–3% of the microbial community by 16S rRNA gene amplicon analysis, and had high metagenomic read recruitment and transcriptional activity in the sediments of Kongsfjord and Van Keulenfjord, Svalbard. This is in line with previous studies that showed Woeseiales is an abundant lineage in marine sediments worldwide [7, 12, 13], although the properties that determine such ecological success are poorly understood. Previous work has suggested metabolic plasticity to be at the root of the global distribution and high abundance continuously observed for Woeseiales worldwide [7, 13, 15]. Our high-resolution transcriptional evidence supports this idea and suggests that the Woeseiales in these Arctic fjords are able to sense and adapt to their environment upon burial, most likely in response to changes in redox chemistry. In addition to metabolic plasticity, transcriptional evidence also supports an additional mechanism that is potentially driving their ecological success: the transcription of a small, membrane-associated stress protein, SP21 (*hspA*) [35, 36].

### Metabolic response to burial

In these surface sediments, where redox cycling of carbon, nitrogen, and sulfur compounds provide key feedbacks to the overlying water [22, 23, 25], we found metabolic potential for chemolithoautotrophy with sulfur oxidation paired to carbon fixation (Fig 3a). This agrees with previous work that hypothesized Woeseiales as members of the sulfur-oxidizing community in marine sediments [7, 12, 13]. We also found strong transcriptional evidence that Woeseiales contribute to denitrification by performing the $NO_2$ reduction step in the examined sediment surface layers down to 5–6 cm (Fig 3b; S1 File). Transcriptional evidence suggests that these metabolisms are occurring *in situ* and suggests sulfur oxidation is down-regulated quickly with depth in response to changing redox conditions (Fig 3b). The electron sources driving nitrite reduction need to be examined in the future. Metabolic plasticity within the Woeseiales was first proposed by Dyksma et al. [13]. Using combined molecular and isotopic approaches, they found support for Woeseiales to switch between sulfur and hydrogen oxidation. They hypothesized that the global distribution and ubiquity of the Woeseiales clade may in part be attributable to their metabolic flexibility. Evidence for the potential for metabolic plasticity was supported by an analysis of metabolic features within all available Woeseiales genomes [7]. Likewise, analysis of MAGs from permeable marine sediments in Australia suggest Woeseiales encode genes for aerobic heterotrophy, denitrification, sulfur oxidation, and hydrogenotrophic sulfur reduction [15]. Further, meta-analysis of deep-sea versus coastal Woeseiales taxa found chemoheterotrophic metabolism is a deep-sea trait [12]. Our analysis extends these findings of metabolic flexibility to include Arctic fjord sediments, suggesting that metabolic flexibility likely drives the ecology and distribution of Woeseiales in the marine sediment realm.

### Dormancy and quiescence strategies

An additional mechanism that may facilitate the worldwide distribution of Woeseiales is its transcriptional response to burial, suggesting cellular quiescence (Fig 4). The extremely high number of transcripts of the single gene encoding SP21, or *hspA* (6–161 times higher than the next most abundant transcript; S7 Fig) that increases with depth (1.5 to 9 times increase over

the upper 5 cm at three sites) introduces a potentially important and overlooked strategy available to Woeseiales. SP21 is transcribed alongside other chaperones and genes for repair, perhaps as a general response to stressful conditions that occur as Woeseiales is buried. While the transcription of chaperones, repair proteins, and other genes related to stress was observed, transcription did not increase with depth and remained several orders of magnitude lower than transcription for spore protein SP21 on average (S9 Fig). Therefore, the transcription of spore protein SP21 may indicate a quiescence strategy related to stress-induced dormancy [44]. The conditions that trigger this response may not be as pronounced or had as much time to develop at site AC, where SP21 transcript abundance remained steady and diversity of transcripts did not decrease with depth. Swath-bathymetry of Van Keulenfjorden indicates that this part of the fjord contains submarine landform assemblages, such as dewatering structures and pockmarks, that result from rapid sediment deposition and debris flows [21]. These structures are absent at site AB. This suggests that that differences in the physical sediment environment, such as rapid deposition, at site AC may attenuate the stressors that elicit SP21 transcription in shallow sediment layers.

In *Stigmatella aurantiaca*, the expression of membrane-localized clusters of SP21 is inducible under stress conditions like heat shock, anoxia, and nutrient deprivation [35, 45]. When expressed under heat shock, antigens for this stress protein have area densities ranging from ~400 to >800 conjugates/$\mu m^2$ within the cell wall [35], suggesting that the protein is in extremely high concentrations in the cell. Unlike the sulfate-reducing Firmicutes whose spores are inducible under high temperatures in Svalbard fjord sediment [46], our transcriptional evidence for putative spore-like dormancy in Woeseiales is limited to just one protein located in the cell wall. Thus, it is very unlikely that Woeseiales can form spores *sensu stricto* as is observed in the Firmicutes, such as *Bacillus subtilis*. This is supported by the inability of the isolate *Woeseia oceani* XK5 to form spores [14]. However, the extremely high transcript abundance of SP21 relative to all other genes, and its sharp increase with depth suggest that it might also have a high protein abundance, as it does in *S. aurantiaca*, where it provides a spore-like quiescence.

Indeed, decreases in abundance and diversity of transcripts for metabolism, information storage and processing, and cellular processes may represent a stress response pattern indicating entry into a stage of quiescence. The transcriptional behavior observed for the Woeseiales MAGs is similar to quiescence strategies noted for other organisms, such as *B. subtilis* and *Mycobacterium tuberculosis* [44]. For example, the overall high transcription of a single protein coupled to much lower overall mRNA diversity and abundance agrees with a recent RNA-seq study on *B. subtilis* spores conducted by Korza et al. [47]. *B. subtilis* spores had overall low transcriptional diversity and abundance, with the exception of a few high-abundance transcripts that were ~4,000-fold more abundant than the least abundant mRNAs [47]. These high-abundance transcripts encoded proteins involved in stringent response and spore coat formation are thought to remain abundant post-sporulation because of mRNA stability conferred by dehydration upon spore maturation [47, 48]. Stabilization of mRNA is also observed for sporulating Actinobacteria [49, 50] and Mycobacteria entering into quiescence [51]. For most sites, mRNA diversity and abundance were highest in uppermost sediment layers prior to build up of SP21 transcripts. This is consistent with increased macromolecular synthesis observed to occur during entry and exit from quiescence in *E. coli* and *M. tuberculosis* [52] in order to facilitate cellular remodeling between growth states [53]. After quiescence is established, transcription has been shown to decrease 3- to 5- fold in *Saccharomyces cerevisiae* [54]. And so, while Woeseiales may not have the ability to form true spores, it may have a strategy that is similar to cellular quiescence that has been observed for both pathogenic bacteria [55] and environmental bacteria [56].

Altogether, high transcription of SP21, coupled to the transcription of chaperones, transcriptional regulators, membrane-associated stress genes, spore coat protein A (S9 Fig), and peptidases related to sporulation factors (M50B), suggests that Woeseiales has a strategy for cellular quiescence for surviving periods of stress. Such metabolic standby of microbial populations, wherein growth and metabolism are extremely depressed or put on hold entirely until conditions again become optimal, has been proposed as a key strategy of life in marine sediments [6, 57]. These dormant populations act as seed banks that can potentially be reactivated under different conditions [58], such as those resulting from resuspension events or bioturbation [59]. Such a subsistence strategy may confer resilience to Woeseiales populations, allowing their persistence in the environment and contributing, in part, to their global distribution and abundance.

## Conclusions

Our results suggest that the Woeseiales MAGs reconstructed from Arctic fjord sediments responded physiologically to burial over the upper 5 cm of sediments. Transcriptional evidence for metabolic plasticity was found with shallow termination of transcription of rDSR likely involved in sulfur oxidation tied to carbon fixation and deeper transcription of the nirS gene for denitrification. A single membrane-associated stress protein, SP21, was highly transcribed at all depths and at all sites. Increases in transcripts for SP21 at three sites coincided with decreased investment in mRNA for metabolism, cellular processes and signaling, and information storage and processing genes with depth. The site where SP21 did not increase with depth did not show decreases in the abundance and diversity of cellular transcripts. Altogether, this suggests that in addition to metabolic flexibility, the success of Woeseiales as a ubiquitous and abundant marine sediment clade may be attributable at least in part to a novel subsistence strategy that involves SP21. With multiple subsistence strategies (metabolic flexibility and cellular quiescence) available to Woeseiales both within and outside the reach of large glacial deposits, they may be well-prepared to withstand shifts in the sub-benthic conditions of Arctic fjords due to climate change.

## Materials and methods

### Sediment collection

Sediment was collected in the summer of 2016 from four different sites within Svalbard fjords (79˚N) with a HAPs corer [60] (Fig 1a). Sediments included in this study are from sites AB (77˚35.249' N, 15˚05.121'E) and AC (77˚32.260' N, 15˚39.434' E) in Van Keulenfjorden (outer and middle sites, respectively; Fig 1b) and sites F (78˚55.075' N, 12˚15.929' E) and P (78˚57.915'N, 12˚15.600'E) in Kongsfjorden (both located at areas of glacier outflow) (Fig 1c).

### Geochemical measurements

Acrylic liners were used for collecting cores from each site only cm apart within each larger HAPs corer. These cores were kept at ~5˚C until they were brought back to land and processed 1–3 days later. Cores were subsectioned every 3 cm down to a depth of ~25 cm. Samples for hydrogen analysis followed a previously published headspace equilibration method [61]. Briefly,1 mL of sediment was placed into a dark glass serum vial which was then crimp sealed and gassed with $N_2$ for 15 min prior to storage at 4˚C. Headspace was measured with glass syringes on a Peak Performer GC with a mercuric chloride detector (San Jose, California) at the University of Tennessee, Knoxville (UTK) after 2 days. Sediment for analysis of organic matter was freeze-dried after thawing from -80˚C and subjected to acid fumigation overnight

before analysis [62]. Total organic carbon (TOC) and isotope composition of carbon ($\delta^{13}C_{org}$) from bulk organic matter was measured using a Thermo-Finnigan Delta XL mass spectrometer coupled to an elemental analyzer at UTK. Carbon to nitrogen (C/N) ratios were calculated by dividing percent C by percent N. Isotopic values were calibrated against the USGS40 and USGS41 international standards. In-house standard sets were run every 12 samples. Across multiple runs, one standard deviation was 0.1–0.2‰ for $\delta^{13}C_{org}$, 1.1–1.8% for mgN, and 1.0–2.2% for mgC. Values for Van Keulenfjorden are published elsewhere [22].

## DNA extraction and 16S rRNA gene amplicon libraries

Cores were collected with acrylic liners a few cm apart from those taken for geochemical analysis and were subsectioned at 1 cm depth intervals in the Ny-Ålesund Marine Lab down to ~20 cm depth. Sediment was frozen immediately on dry ice and remained frozen during transport using a dry shipper. Sediment was stored at -80˚C until processing. Nucleic acids were extracted by the Lloyd Lab (sites AB, AC, P, F) using the Qiagen RNeasy Powersoil® Kit (Hilden, Germany) for RNA with the DNA extraction accessory according to manufacturer's instructions. DNA from site AC at 18 cm depth was extracted by the Loy lab using the DNeasy PowerSoil Kit (Qiagen), following manufacturer's instructions. 16S rRNA gene amplification from sites P and F in Kongsfjorden failed. The Phusion Master Mix (Thermo Fisher) was used with the primer set 515F/806R [63] at the Center for Environmental Biology at UTK for amplification. Reads were sequenced with Illumina MiSeq and trimmed for quality with Trimmomatic [64] using a window 10 base pairs wide and a minimum phred score of 28. Trimmed reads were then processed in mothur 1.35.1 [65] using the computational cluster at the Bioinformatics Resource Facility (BRC) at UTK. Operational taxonomic units (OTUs) were clustered with mothur as previously described [22]. All 16S rRNA gene amplicon library sequences can be accessed at NCBI under Bioproject PRJNA493859.

## Metatranscriptomic and metagenomic sequencing

RNA extracts were treated with DNAse I at UTK (Qiagen) and further DNase treatment was performed at MRDNA (Shallowater, TX), followed by sequencing of metatranscriptomic libraries with Illumina HiSeq, PE 2x250 bp. Individual 1 cm-depth resolved metatranscriptomic libraries were generated with RNA extracts from the first 5 to 6 cm of sediment from sites AB, AC, F, and P, for a total of 20 metatranscriptomes. Metagenomic libraries were generated from the combined extracts from the first 5 cm (spanning 0 to 5 cm downcore) in sites AB and F in the Center for Environmental Biotechnology, Knoxville, using Illumina HiSeq, PE 2x250 bp. The decision to make high depth resolution metatranscriptomes, and only a single metagenome for each site was based on financial constraints. The Loy Lab sequenced DNA extracts from Site AC at 18 cm depth using Illumina HiSeq 3000, PE 2x150 by the Biomedical Sequencing Facility, at The University of Vienna.

Prior to metagenomic assembly, raw reads were trimmed for quality and adapters were removed using in-house scripts in the Loy lab (removing the leading eight 5' bases, bases with QC < 15 and reads below 50 bp in length) and Illumina adapters were removed using AdapterRemoval using default settings [66]. The Lloyd Lab used Trimmomatic [64] for trimming both metagenomes and metatranscriptomes, with a sliding window of 10 and a phred cut off score of 28 for all reads above 90 bp. The quality of trimmed reads was assessed with Quast 4.5 [67]. Metagenomic sequences can be accessed in the NCBI SRA under BioSamples SAMN10372305, SAMN10372307, and SAMN10372306.

## Metagenomic assembly: MetaSPAdes

Metagenomic assembly was accomplished using metaSPAdes [68] both in command line for Linux and in KBase, a browser interface with bioinformatics modules and applications [69]. Site F was the only metagenome that was assembled via metaSPAdes version 3.11 in the command line, with kmer size set to 21, 33, 55, 77, 99, and 127 to find the best assembly. All other metaSPAdes assemblies were completed on KBase with the default parameters (1000 bp length and kmer sizes of 21, 33, and 55). Assembled contigs were then filtered to contain only contigs with more than 5x coverage and 1000 bp length using in-house scripts.

## Metagenomic assembly: IDBA and Megahit

To reduce RAM utilization and wall clock time, larger sequence datasets were normalized to 100X with BBN prior to assembly with IDBA version 1.1.3 [70] and Megahit version 1.0.5 [71]. Assemblies were completed in either the command line or on KBase [69]. All assemblies were generated with 1000 minimum contig length, except IDBA assembly in the Loy lab, which used a 500 bp cutoff.

## Taxonomic binning of contigs into metagenome assembled genome (MAGs)

Contig binning was carried out with MaxBin2 version 2.2.3 [72], CONCOCT (version 0.4.1) [73] and MetaBAT (version 2.12.1) [74] in command line. Each of these binning tools utilizes genomic signatures within contigs, such as coverage and tetranucleotide frequency, to identify discrete clusters of contigs that likely represent a population's genome. MaxBin2 and CONCOCT binning were performed with default parameters and 1000 bp minimum contig length. MetaBat binning was conducted with an interactive pipeline that applies decreasing levels of stringency within each successive iteration, collecting the best bins and their contigs each time. The remaining contigs are then placed into the next step with a lower stringency.

DASTool (version 1.1.0) [75] was applied to each set of binned contigs to create medium to high quality MAGs. This tool takes several different MAGs as input and identifies a consensus, non-redundant genome for each MAG, leading to higher quality genomes. Quality was determined for our MAGs according to Bowers et al. [76]. Medium quality drafts have completeness and contamination values of $\geq$ 50% and <10%, respectively whereas high quality MAGs have completeness and contamination values >90% and contamination <5%. Completeness and contamination were determined with CheckM [77]. MAG taxonomy was determined through phylogenetic analysis using a concatenated alignment of proteins from single copy marker genes included in the CheckM suite and a tree was built with FastTree [78].

## Transcript mapping and annotation

Ribosomal sequences were identified and removed using SortmeRNA version 2.1 [79]. Filtered transcripts were mapped to MAGs using Bowtie2 version 2.3.4 [80] with sensitive local mapping, which requires neither end-to-end alignment nor mismatches when evaluating seed substrings 20bp in length. Genomes were annotated with Prokka version 1.12 [81]. Mapping files, Prokka gene calls, and fasta files for each MAG were then used in the metagenomics pipeline of Anvi'o version 5.1 [82] for COG identification, visualization, and transcript abundance estimates. Transcript abundance is the coverage of the gene normalized by gene length. Transcripts per million was calculated by dividing transcript abundance by the number of mapped reads for each library and multiplying by 1,000,000.

COG annotations were computed using eggNOG-mapper [83] based on eggNOG 4.5 ontology data [84]. Spearman correlation coefficient analysis of changes in transcriptional richness associated with depth was performed in R [85]. Intersection analysis was performed with the UpsetR package in R [86]. TPM values for each *hspA* gene copy were log-transformed and normality was tested with a shapiro-wilks test in R [86]. ANVOA (alpha = 0.05) was performed in R to facilitate post-hoc Tukey testing with the agricolae package (alpha = 0.05). Co-localization analysis was performed in CLC Genomics Workbench.

Amino acid identity analysis for each Woeseiales MAG was conducted with AAI-profiler [87]. This tool uses SANSparallel [88] to find homology matches within the Uniprot database (update Apr2020).

## RubisCO, rDSR, and hspA sequence analyses

Publicly available full length (> 480 aa) sequences for large and small chains of ribulose bisphosphate carboxylase (RubisCO) were compared with MAG RubisCO sequences. Mega version 7 [89] was used for Clustal alignments and neighbor-joining tree construction.

The alpha subunits of Woeseiales rDSR sequences were aligned in the *dsr*AB ARB database [90]. RAxML was used within the CIPRES Science Gateway v. 3.3 server [91] and the Interactive Tree of Life (iToL) server was used for tree visualization [92]. For spore protein SP21, publicly available amino acid sequences for *hsp*A were acquired from environmental samples and isolates and compared with Woeseiales sequences. RAxML was used within the CIPRES Science Gateway v. 3.3 server [91] and the Interactive Tree of Life (iToL) server was used for tree visualization [92].

## Supporting information

**S1 File.**
(XLSX)

**S1 Table. Genome statistics for the Woeseiales MAGs in this study.** Completeness and contamination were determined for each genome with CheckM.
(DOCX)

**S2 Table. Genes with transcripts detected at all sites. C** = Energy production and conversion; **D** = Cell cycle control, cell division, chromosome partitioning; **E** = Amino acid transport and metabolism; **F** = Nucleotide transport and metabolism; **G** = Carbohydrate transport and metabolism; **H** = Coenzyme transport and metabolism; **I** = Lipid transport and metabolism; **J** = Translation, ribosomal structure and biogenesis; **K** = Transcription; **L** = Replication, recombination and repair; **N** = Cell motility; **O** = Post-translational modification, protein turnover and chaperones; **P** = Inorganic ion transport and metabolism; **Q** = Secondary metabolites biosynthesis, transport and catabolism; **S** = Function unknown; **T** = Signal transduction mechanisms; **U** = Intracellular trafficking, secretion and vesicular transport; **V** = Defense mechanisms.
(DOCX)

**S1 Fig. Phylogenetic tree for Woesiales MAGs placement.** Maximum likelihood was calculated in Mega v. 7 with 1000 bootstraps on a concatenated alignment of 49 ribosomal proteins. Only nodes with >75% support are shown. MAGs from this study are indicated with blue circles. Tree was rooted using Roseobacter species as the outgroup.
(TIF)

**S2 Fig. Comparison of genomic contents across the 5 Woeseiales MAGs to show similarity.**
Figure generated with KEGG-decoder [26].
(TIF)

**S3 Fig. Amino acid identity analysis.** Scatterplots, Krona diagrams, and histograms relating
amino acid identity to the cultured isolate, *Woeseia oceani*, are presented for the MAGs Woe-
seia_stnAB (a, b, c), Woeseia_stnAC (d, e, f), Woeseia2_stnAC (g, h, i), Woeseia_stnF (j, k, l),
and Woeseia2_stnF (m, n, o).
(TIF)

**S4 Fig. Intersection analysis of all transcripts.** Set intersections of transcripts visualized with
UpsetR [86]. The total number transcribed genes ("set size") is indicated as horizontal bars.
Vertical bars represent the intersection size, or number of transcribed genes that fall into the
15 different sets of site combinations. Depth analysis of transcript abundance for unique sets
of transcripts are depicted in S5 Fig.
(TIF)

**S5 Fig. Recruitment of unique transcripts.** Transcriptional coverage (transcript abundance
normalized to gene length) is reported for the unique transcripts detected at each site for the
COGs Cellular Processing and Signaling, Information Storage and Processing, Metabolism,
and Function Unknown. Those with predicted function only were removed from this analysis.
Dot size scales with metagenome coverage and color saturation scales from lighter to darker
blue for increased transcriptional coverage. Site P did not have a metagenome.
(TIF)

**S6 Fig. Neighbor-joining tree of RubisCO sequences.** Only nodes with >80% support after
1000 bootstraps are shown.
(TIF)

**S7 Fig. Coverage of all Woeseiales MAG transcripts.** The transcripts per million (TPM)
value is reported for each transcribed gene (along x-axis). Names of genes are not included to
ease illustration and interpretation. COG assignment of genes is represented by color. The
gene encoding for spore protein SP21 is indicated with arrows.
(TIF)

**S8 Fig. Maximum likelihood tree of *hsp*A genes encoding for spore protein SP21.** Bootstrap
support after 1000 bootstraps is reported for nodes with > 70% support. Woeseiales sequences
are indicated in red, cultured microorganisms in purple, and experimentally-verified spore
protein SP21 indicated in green [36, 45]. Archaeal sequences of spore protein SP21 were used
as the outgroup. The scale bar represents the number of substitutions per site.
(TIF)

**S9 Fig. Coverage with depth for transcripts of stress proteins, chaperones, and transcrip-
tional regulators.** TPM values reported for genes associated with starvation response, stress
mitigation, protein repair/folding, and transcriptional regulation. Yellow border labels indicate
genes that co-occur on the same contigs as *hsp*A (encoding spore protein SP21; S9 Fig).
(TIF)

**S10 Fig. Co-localization of genes on the same contig as the gene for spore protein SP21
(*hsp*A), highlighted with orange rectangle.** Visualizations from CLC Genomics Workbench.
(TIF)

## Acknowledgments

Computing resources were provided by the Center for Dark Energy Biosphere Investigations (C-DEBI, publication number 539). We thank Dr. B. Tully for assistance with computer programming. We also thank Captain Stig Henningsen and first mate Reidar Sorensen of MS Farm. We thank the Alfred Wegener Institute—Institute Paul Emile Victor (AWIPEV) station and staff for housing and excellent logistics support. We thank all the participants of the 2016 Svalbard KOP 56/RiS 10528 expedition for help with sample collection.

## Author Contributions

**Conceptualization:** Joy Buongiorno, Karen G. Lloyd.

**Data curation:** Joy Buongiorno, Kenneth Wasmund.

**Formal analysis:** Joy Buongiorno.

**Funding acquisition:** Joy Buongiorno, Alexander Loy, Karen G. Lloyd.

**Investigation:** Joy Buongiorno, Katie Sipes.

**Methodology:** Joy Buongiorno, Kenneth Wasmund, Karen G. Lloyd.

**Project administration:** Karen G. Lloyd.

**Resources:** Kenneth Wasmund, Alexander Loy, Karen G. Lloyd.

**Software:** Kenneth Wasmund, Alexander Loy.

**Supervision:** Alexander Loy.

**Visualization:** Joy Buongiorno.

**Writing – original draft:** Joy Buongiorno.

**Writing – review & editing:** Joy Buongiorno, Katie Sipes, Kenneth Wasmund, Alexander Loy, Karen G. Lloyd.

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
