## [Decision Letter · Decision Letter 0]

9 Jul 2020

PONE-D-20-16591

Woeseiales transcriptional response in Arctic fjord surface sediment

PLOS ONE

Dear Dr. Buongiorno,

Thank you for submitting your manuscript to PLOS ONE. After careful consideration, we feel that it has merit but does not fully meet PLOS ONE’s publication criteria as it currently stands. Therefore, we invite you to submit a revised version of the manuscript that addresses the points raised during the review process.

We look forward to receiving your revised manuscript.

Kind regards,

Peter R. Girguis, Ph.D.

Academic Editor

PLOS ONE

Journal Requirements:

2. We note that Figure 1 in your submission contain map images which may be copyrighted. All PLOS content is published under the Creative Commons Attribution License (CC BY 4.0), which means that the manuscript, images, and Supporting Information files will be freely available online, and any third party is permitted to access, download, copy, distribute, and use these materials in any way, even commercially, with proper attribution. For these reasons, we cannot publish previously copyrighted maps or satellite images created using proprietary data, such as Google software (Google Maps, Street View, and Earth). For more information, see our copyright guidelines: http://journals.plos.org/plosone/s/licenses-and-copyright.

2.1. You may seek permission from the original copyright holder of Figure 1 to publish the content specifically under the CC BY 4.0 license. 

2.2. If you are unable to obtain permission from the original copyright holder to publish these figures under the CC BY 4.0 license or if the copyright holder’s requirements are incompatible with the CC BY 4.0 license, please either i) remove the figure or ii) supply a replacement figure that complies with the CC BY 4.0 license. Please check copyright information on all replacement figures and update the figure caption with source information. If applicable, please specify in the figure caption text when a figure is similar but not identical to the original image and is therefore for illustrative purposes only.

Reviewers' comments:

Reviewer's Responses to Questions

**Comments to the Author**

1. Is the manuscript technically sound, and do the data support the conclusions?

Reviewer #1: Yes

Reviewer #2: Yes

2. Has the statistical analysis been performed appropriately and rigorously? 

Reviewer #1: Yes

Reviewer #2: Yes

3. Have the authors made all data underlying the findings in their manuscript fully available?

Reviewer #1: Yes

Reviewer #2: Yes

4. Is the manuscript presented in an intelligible fashion and written in standard English?

Reviewer #1: Yes

Reviewer #2: Yes

5. Review Comments to the Author

Reviewer #1: This a well designed, performed, and written study tracking the genomic diversity and metabolic activity of bacteria through fjord sediments. I have no issues with it in its current form and think it should be published.

Reviewer #2: This manuscript examines five novel metagenome assembled genomes (MAGs) from the Order Woeseiales, a bacterial clade which is abundant in marine sediments, coupled with depth-resolved metatranscriptomic data, from four sampling sites in Arctic fjord surface sediment. The authors report that these MAGs had the capability for chemoautotrophic carbon fixation via the Calvin-Benson-Bassham pathway, coupled to the oxidation of sulfite or thiosulfate, as well as the ability to reduce nitrite to nitric oxide. At three of the four sampling sites, transcription of most of the central metabolism genes dropped with sediment depth, while the transcription of a single gene, spore protein SP21, rose sharply. The authors propose that the Woeseiales in this environment may enter a quiescent state with burial, allowing these microbes to ride out stressful conditions by entering dormancy, only to be revived when improved conditions present themselves (if, for example, they are brought to the surface of the sediment through bioturbation).

While it is risky to extrapolate data from five MAGs to describe the metabolic capabilities of an entire clade, the power of the data presented in this manuscript lies in the coupling of the data to the metatranscriptomes. In this way the authors demonstrate how the Woeseiales found in their samples might survive changing conditions with sediment burial.

The science presented here is well-executed and the manuscript well-written. The methods are logical, well-explained, and justified. I found very little to criticize, but I do have a few minor points:

The authors state towards the end of the discussion that SP21 increased with depth at three of the four sampling sites where they were able to generate metagenomes/metatranscriptomes, and that at the one site where they did not observe this, they also did not observe a concurrent decrease in the abundance and diversity of cellular transcripts. (I assume from the figures that the outlier site is site AC, although this is not stated in the text.) Do the authors have any insight into why this site may be different from the others? Could there be some geochemical parameter at play here? There may of course be no obvious explanation, but a comment or two about what may be driving this difference would be useful.

Line 454 (methods): "extracted by both the Lloyd" is a phrase missing here? Do you mean "both the Lloyd and the Loy labs"?

Figure 5: Some of the depths appear to be missing from sites AB and AC (unless I misinterpreted something in the text). The figure legend mentions that certain libraries from site F were not included- at first it was difficult to tell that points were missing from the line graph at the bottom of the figure, but I can see that those are not there. What happened with the data from sites AB and AC?

6. PLOS authors have the option to publish the peer review history of their article (what does this mean?). If published, this will include your full peer review and any attached files.

Reviewer #1: No

Reviewer #2: **Yes: **Lauren M Seyler

---

## [Author Response · Author response to Decision Letter 0]

15 Jul 2020

We have re-made Figure 1 to include only images from NASA Earth Observatory. We have changed the figure caption to reflect the proper attribution of image data source. 

Reviewer comments:

Reviewer #1: This a well designed, performed, and written study tracking the genomic diversity and metabolic activity of bacteria through fjord sediments. I have no issues with it in its current form and think it should be published.

The authors thank this reviewer for taking the time to reviewer our work and for their positive review.

Reviewer #2: This manuscript examines five novel metagenome assembled genomes (MAGs) from the Order Woeseiales, a bacterial clade which is abundant in marine sediments, coupled with depth-resolved metatranscriptomic data, from four sampling sites in Arctic fjord surface sediment. The authors report that these MAGs had the capability for chemoautotrophic carbon fixation via the Calvin-Benson-Bassham pathway, coupled to the oxidation of sulfite or thiosulfate, as well as the ability to reduce nitrite to nitric oxide. At three of the four sampling sites, transcription of most of the central metabolism genes dropped with sediment depth, while the transcription of a single gene, spore protein SP21, rose sharply. The authors propose that the Woeseiales in this environment may enter a quiescent state with burial, allowing these microbes to ride out stressful conditions by entering dormancy, only to be revived when improved conditions present themselves (if, for example, they are brought to the surface of the sediment through bioturbation).

While it is risky to extrapolate data from five MAGs to describe the metabolic capabilities of an entire clade, the power of the data presented in this manuscript lies in the coupling of the data to the metatranscriptomes. In this way the authors demonstrate how the Woeseiales found in their samples might survive changing conditions with sediment burial.

The science presented here is well-executed and the manuscript well-written. The methods are logical, well-explained, and justified. I found very little to criticize, but I do have a few minor points:

 The authors state towards the end of the discussion that SP21 increased with depth at three of the four sampling sites where they were able to generate metagenomes/metatranscriptomes, and that at the one site where they did not observe this, they also did not observe a concurrent decrease in the abundance and diversity of cellular transcripts. (I assume from the figures that the outlier site is site AC, although this is not stated in the text.) Do the authors have any insight into why this site may be different from the others? Could there be some geochemical parameter at play here? There may of course be no obvious explanation, but a comment or two about what may be driving this difference would be useful.

Thank you for inviting us to speculate about this. We have added text (L376-383) that addresses a potential cause for the differences observed at site AC.

Line 454 (methods): "extracted by both the Lloyd" is a phrase missing here? Do you mean "both the Lloyd and the Loy labs"?

 We have fixed this in the text. (The Lloyd Lab extracted DNA and RNA using the RNeasy kit with DNA accessory kit and the Loy lab extracted DNA using the DNeasy kit). 

Figure 5: Some of the depths appear to be missing from sites AB and AC (unless I misinterpreted something in the text). The figure legend mentions that certain libraries from site F were not included- at first it was difficult to tell that points were missing from the line graph at the bottom of the figure, but I can see that those are not there. What happened with the data from sites AB and AC?

Thanks for checking this. All libraries are reported in the original figure, it just looks as though some are missing as you suggest because other sites had successful metatranscriptomes at different depths.

Site AB had four metatranscriptomes (0.5, 1.5, 2.5, 3.5 cmbsf)

Site AC had four metatranscriptomes (1.5, 2.5, 3.5 4.5 cmbsf)

---

## [Editor Report · Decision Letter 1]

5 Aug 2020

Woeseiales transcriptional response to shallow burial in Arctic fjord surface sediment

PONE-D-20-16591R1

Dear Dr. Buongiorno,

We’re pleased to inform you that your manuscript has been judged scientifically suitable for publication and will be formally accepted for publication once it meets all outstanding technical requirements.

Kind regards,

Peter R. Girguis, Ph.D.

Academic Editor

PLOS ONE
---

## [Editor Report · Acceptance letter]

11 Aug 2020

PONE-D-20-16591R1 

Woeseiales transcriptional response to shallow burial in Arctic fjord surface sediment 

Dear Dr. Buongiorno:

I'm pleased to inform you that your manuscript has been deemed suitable for publication in PLOS ONE. Congratulations! Your manuscript is now with our production department. 

Kind regards, 

on behalf of

Dr. Peter R. Girguis 

Academic Editor

PLOS ONE